# Where there is no nurse: an observational study of large-scale mentoring of auxiliary nurses to improve quality of care during childbirth at primary health centres in India

Krishna D Rao,[ ][1] Swati Srivastava,[1,2] Nicole Warren,[3] Kaveri Mayra,[4] Aboli Gore,[5] Aritra Das,[5] Saifuddin Ahmed[6]

For numbered affiliations see end of article.

**Correspondence to**
Dr Krishna D Rao;
kdrao@jhu.edu

## ABSTRACT

**Objective** Clinician scarcity in Low and Middle-Income Countries (LMIC) often results in de facto task shifting; this raises concerns about the quality of care. This study examines if a long-term mentoring programme improved the ability of auxiliary nurse-midwives (ANMs), who function as paramedical community health workers, to provide quality care during childbirth, and how they compared with staff nurses.

**Design** Quasi-experimental post-test with matched comparison group.

**Setting** Primary health centres (PHC) in the state of Bihar, India; a total of 239 PHCs surveyed and matched analysis based on 190 (134 intervention and 56 comparison) facilities.

**Participants** Analysis based on 335 ANMs (237 mentored and 98 comparison) and 42 staff nurses (28 mentored and 14 comparison).

**Intervention** Mentoring for a duration of 6–9 months focused on nurses at PHCs to improve the quality of basic emergency obstetric and newborn care.

**Primary outcome measures** Nurse ability to provide correct actions in managing cases of normal delivery, postpartum haemorrhage and neonatal resuscitation assessed using a combination of clinical vignettes and Objective Structured Clinical Examinations.

**Results** Mentoring increased correct actions taken by ANMs to manage normal deliveries by 17.5 (95% CI 14.8 to 20.2), postpartum haemorrhage by 25.9 (95% CI 22.4 to 29.4) and neonatal resuscitation 28.4 (95% CI 23.2 to 33.7) percentage points. There was no significant difference between the average ability of mentored ANMs and staff nurses. However, they provided only half the required correct actions. There was substantial variation in ability; 41% of nurses for normal delivery, 60% for postpartum haemorrhage and 45% for neonatal resuscitation provided less than half the correct actions. Ability declined with time after mentoring was completed.

**Discussion** Mentoring improved the ability of ANMs to levels comparable with trained nurses. However, only some mentored nurses have the ability to conduct quality

### Strengths and limitations of this study

► We were able to assess if the quality of obstetric care produced by auxiliary nurse-midwives (ANMs), who typically serve as paramedical workers in rural communities in India, can be improved through on-site mentoring to levels comparable with trained staff nurses.

► This was a large scale on-site mentoring programme that was implemented in around 340 (60%) primary care facilities and included both ANMs and staff nurses in the state of Bihar.

► Nurse ability measured using a unique tool that captures both knowledge and demonstrated skill.

► Observational study includes both intervention (mentored) and comparison (non-mentored) ANMs; however, no preintervention observations on nurse ability, but staff nurses serve as reference standard.

► ANM and staff nurse demonstrated ability to provide quality care can be different from what they do in actual practice.

deliveries. Continuing education programmes are critical to sustain quality gains.

## INTRODUCTION

Access to a skilled birth attendant is a key strategy to reduce maternal and neonatal deaths in low/middle-income countries. India's high maternal mortality of 130 per 100 000 live births contributes 19% of global maternal deaths.[1 2] Institutional deliveries have been actively promoted by the government to increase access to skilled birth attendants; for example, since 2005 the government has offered cash incentives to pregnant women (and health workers at public facilities) if they deliver at a health

facility. This has resulted in the institutional delivery rate doubling between 2005 and 2014 (from 39% to 79%).[3] Notably, around 52% (54% in rural) of the deliveries nationwide occurred at a public sector facility; this highlights the prominent role of the public sector in providing women access to skilled birth attendants.[3]

The remarkable increases in institutional deliveries in India have also raised concern about the quality of obstetric care. Studies have pointed out that despite substantial increases in institutional deliveries there has not been a concomitant reduction in maternal mortality, and its effect on neonatal mortality is unclear.[4–10] While the infrastructure and resources available at health facilities have been an ongoing challenge, more recently, attention has also focused on the performance of healthcare providers.[11–14] A recent study of nurses' management of obstetric complications at public facilities in India found that only 14% of nurses conducted initial assessments, 58% made a correct clinical diagnosis and 20% provided first-line care.[14] A variety of factors constrain the ability of nurses to provide quality care—inadequate infrastructure, poor training, limited access to continuing education and professional support from experienced clinicians, lack of agency in the workplace and entrenched detrimental clinical practices.[6 15–18]

In health system contexts where trained clinicians are scarce, clinical functions are often shifted to health workers with less training and without appropriate support, which raises concerns about the quality of care.[19] In the state of Bihar, where this study is situated, the lack of doctors and staff nurses at primary health centres (PHCs) has resulted in lesser trained auxiliary nurse-midwives (ANMs) providing obstetric care. Both staff nurses and ANMs are cadres within the public sector health system. Staff nurses have finished high school and have completed a diploma in General Nursing and Midwifery (GNM), a 3-year course in basic nursing.[20] GNM nurses are trained to conduct deliveries and manage complications. ANMs have completed high school and possess a 2-year diploma that trains them to be multipurpose community health workers; their course work includes some training in managing deliveries.[20] Discussions the authors had with ANMs in Bihar indicate that their diploma programmes include some course work but no practical training in managing deliveries. The curriculum of GNM and ANM programmes is set by the Nursing Council of India.[20] ANMs were introduced into the health workforce in the 1950s to serve primarily as midwives at PHCs and subcentres.[21 22] However, overtime their function changed from that of a midwife to a paramedical health worker who provides a range of services such as family planning, immunisations and some level of antenatal care.[21 22] In Bihar, because staff nurses are scarce at PHCs, ANMs have taken on the responsibility of managing deliveries.

Efforts to improve clinical skills of in-service nurses (We use the term nurses to include both ANMs and staff nurses) have traditionally taken the form of short duration training sessions, often away from the health facility, which are not very effective for skill building. On-site clinical mentoring programmes, which involve longer term engagement and relationship building between mentor and mentee, have shown to be effective in improving nurse knowledge and skills around essential obstetric and neonatal care in other countries, and more recently in India.[23–27] When mentors become part of the mentee's environment, mentoring functions such as didactic instruction, demonstrations and support and encouragement, have the potential to improve mentee skills and self-efficacy to perform effectively and overcome challenges. On-site mentoring programmes for nurses at PHCs in Bihar report increased nurse knowledge and management of normal and complicated deliveries, increased adherence to quality protocols during delivery and increased confidence.[26 27] In other contexts, mentoring was associated with a reduction in mortality for low birthweight babies.[28]

This paper reports on a large-scale mentoring programme to improve the quality of intrapartum care produced by ANMs (and staff nurses) at PHCs in the state of Bihar, India. Specifically, this study investigates if mentoring improved the ability of ANMs to manage normal and complicated deliveries; how mentored ANMs and staff nurses compared in ability, and the sustainability of the mentoring effect. Nurse ability was assessed in managing standardised cases of normal delivery, postpartum haemorrhage and neonatal resuscitation. These cases were selected because mentees were taught to manage these conditions; haemorrhage is a leading cause of maternal mortality, and birth asphyxia a leading cause of neonatal mortality in India.[29 30]

The state of Bihar (population 100 million) is among India's resource poor states. PHCs provide outpatient and occasionally inpatient care, maternal health services, family planning, public health services and basic diagnostic and laboratory services. They cover between 20 000 and 30 000 people and are staffed with a doctor, nurses and other supporting staff. PHCs are central to facility-based delivery of obstetric services because they are the first level in the public system which has clinicians trained to provide obstetric care such as doctors and nurses. In 2015–2016, ~64% of all births were in health facilities and 36% at home; urban areas had 74% institutional births while rural areas had 63%.[31] Institutional births in public facilities have increased from just 4% of all institutional births in 2005–2006 to almost 48% in 2015–2016.[31]

### AMANAT nurse mentoring programme in Bihar

CARE-India in collaboration with the Government of Bihar implemented the AMANAT on-site nurse mentoring programme between March 2015 and January 2017. AMANAT aimed to improve the quality of basic emergency obstetric and newborn care (BEmONC) at primary healthcare facilities in the state of Bihar.[32 33] Because the nomenclature of primary care facilities is somewhat ambiguous in Bihar and change over time, we refer to AMANAT health facilities as PHCs in this study. The AMANAT

programme was preceded by a pilot from 2011 to 2013 in eight districts of Bihar.[34] The AMANAT programme was implemented in a staggered manner over four phases; the programme content was similar across phases.[33] In each phase, 80 PHCs were purposively selected by CARE-India for mentoring. Several criteria determined selection of a PHC—availability of nurses or ANMs at the PHCs, the volume of deliveries, the infrastructural readiness of the PHC, the willingness of the PHC management to undertake mentoring and the proximity of the PHC to other PHCs in the area such that mobile mentoring teams could easily rotate between them. In our analysis, we include many of these selection criteria to construct a counterfactual set of health facilities. In total, 400 out of 534 block-level (the smallest administrative unit within a district) PHCs in Bihar received the nurse mentoring intervention.

Prior to mentoring commencing at a PHC, ANMs and staff nurses affiliated with the particular PHC who were willing to participate in the mentoring were identified by the Medical Officer in-charge and requested to be present for the duration of mentoring. On average, there were six to eight ANMs/staff nurses mentored in a PHC. In the AMANAT programme, the same pair of nurse–mentors rotated multiple times through four PHCs over a 6-month to 9-month period, spending one consecutive week at a PHC when they visited (with an average of four PHCs for each nurse–mentor pair).[32] The nurse-mentors were hired by CARE-Bihar, were recruited from across India and had a BSc degree in nursing. The nurse–mentors were trained by CARE and provided with 6 days of training on simulation facilitation, team building, communication skills and debriefing skills, followed by a 4-day refresher training around 3 months into the mentoring period.[35]

During their visit to the PHC, the nurse–mentors used structured learning sessions to provide didactic instruction covering a range of topics and bedside mentoring related to managing normal and complicated deliveries.[32] Nurses were trained on basic nursing procedures, infection prevention, basic obstetric and neonatal practices, management of complications such as postpartum haemorrhage, birth asphyxia, pre-eclampsia and others, documentation and reporting, team rapport and communication.[32 33 35] The typical sequence of mentoring sessions was—team building in the first week, followed by week long sessions on normal delivery and immediate newborn care, neonatal resuscitation and postpartum haemorrhage, and repeated mentoring on these conditions along with other obstetric and neonatal emergencies in the remaining weeks.[33] Tailored, structured simulations were used for normal deliveries and complications management through the PRONTOPack and other simulation kits.[35–37] These kits include MamaNatalie, a birth simulator worn by a demonstrator to resemble a pregnant woman and used for obstetric practices, and NeoNatalie, a neonate model used to demonstrate neonatal practices. Importantly, the nurse–mentors worked alongside mentees, observed them and provided instruction by demonstration through the comanagement of cases. All mentoring and training materials were in the local Hindi language. Mentoring sessions were conducted in groups. ANMs and staff nurses were not mentored in separate groups. While a set of standard topics were covered during mentoring, certain aspects were emphasised to suit the particular learning needs of mentees and/or respond to clinical scenarios that emerged at the mentoring site.

## METHODS
### Study design
To assess the effect of mentoring on nurse ability, we use a quasi-experimental post-test with matched comparison group design. PHCs exposed to AMANAT mentoring were matched on several criteria with non-mentored PHCs (see Analytical methods section). The AMANAT programme was implemented in four phases, such that, in each phase 80 PHCs were purposively selected. Because the nomenclature of primary care facilities is somewhat ambiguous and can include subdistrict hospitals and other types of primary healthcare facilities, and these designations changes over time, we refer to AMANAT health facilities as PHCs in this study. Of the ~534 PHCs in Bihar, 80 PHCs were exposed to mentoring in a pilot phase (2011–2013) and a further 320 were subsequently exposed during the AMANAT programme (2015–2016). Therefore, around 134 PHCs were not exposed to mentoring. This study focuses on the 240 PHCs in phases II, III and IV. These PHCs were first observed twice via repeat cross-sectional surveys—first, just after mentoring was completed and a second survey some months afterwards, depending on the phase. We excluded phase I PHCs because they had completed mentoring ~2 years prior to the survey date and we expected significant number of transfers of mentored nurses during this time.

### Selection of PHCs
The 240 PHCs in phase II, III and IV of the AMANAT programme served as our sampling frame. For phases II and III, which began mentoring at approximately the same time, 40 PHCs from each phase were randomly sampled and a total of 79 out of these 80 selected PHCs were surveyed (One facility in phase II and III received the intervention in the initial pilot phase between January 2011 and December 2013 and was excluded). From phase IV, 80 PHCS were selected and all were surveyed. Thus, a total of 159 mentored PHCs constitute the pool of intervention group PHCs.

One hundred and thirty-four PHCs in Bihar did not receive any mentoring either through the pilot or AMANAT programme. Seven of these PHCs reported 15 or less deliveries per month in the state Health Management Information System (HMIS) and were excluded. The remaining 127 PHCs were mapped to assess geographical proximity to mentored facilities. From these PHCs, 80 PHCs that were closest to a mentored PHC—either from the same block or adjacent block as an intervention

PHC—were purposively selected. These 80 PHCs form the pool of comparison group PHCs.

## Selection of nurses

In the intervention PHCs, only ANMs and staff nurses who had completed mentoring were eligible to participate. For each sampled PHC, the mentored staff nurses or ANMs were identified from mentoring rosters. From this list, two were randomly selected and were requested to be present on the day the survey team arrived. From the 159 mentored PHCs, a total of 314 staff nurses and ANMs were interviewed (out of a target of 318). From the comparison group of PHCs, a convenience sample of two nurses were selected from among those present on the day the survey team arrived. From the 80 comparison PHCs 160 staff nurses and ANMs were interviewed (out of a target of 160).

## Data collection and questionnaires

Between September 2016 and November 2017, a series of cross-sectional surveys of PHCs was conducted. PHCs and nurses in phases II, III and IV were first observed soon after mentoring was completed and the followed-up once after that. Phase II PHCs were visited 3 and 15 months after mentoring was completed, phase III PHCs were visited 1 and 13 months after mentoring was completed and phase IV PHCs were visited just after and 6 months after mentoring was completed. Data were collected using tablets by trained enumerators who had a GNM or higher nursing degree. Enumerators were recruited from other parts of India and trained in Bihar for 10 days to 2 weeks, depending on the particular batch of enumerators, on practices of managing normal and complicated deliveries, and administering the questionnaires.

Nurse ability was assessed using a structured questionnaire that captured both their knowledge and demonstrated skill in managing normal and complicated deliveries (see Analytical methods section). This questionnaire also included information on their background (age, number of months of AMANAT training completed, ANM or GNM nurse, regular or contract staff). Further, data from a cross-section survey of health facilities were carried out on the service delivery readiness of PHCs in Bihar and were used for matching intervention and comparison PHCs. In the PHCs, surveyed information was collected on availability and condition of drugs, supplies, equipment and building condition.

## Patient and public involvement

We contacted four senior nurses in Bihar and elsewhere to guide development of the protocols and to train data collectors. The data collectors were also trained nurses. Feedback from nurses during pre-testing enabled designing study protocols so that they did not to place undue time burden on respondent nurses and were flexible to demands of their professional work.

## Analytical methods

### Matching health facilities

The intervention PHCs were selected on the basis of several criteria (see Introduction section). As such, there are likely to be systematic differences in these and other characteristics between intervention and comparison PHCs. Such observed and unobserved group differences can bias estimates of mentoring impact if they are related to the outcome of interest, that is, nurse ability. We adjust for this potential bias by matching intervention and comparison PHCs on several criteria. The health facility survey collected information on the service delivery readiness of PHCs in Bihar. Using the service delivery/readiness domains indicated in the WHO's Service Availability and Readiness Assessment[38] as a guide, we identified the following six indicators for matching PHCs: location—connected to metaled road; basic amenities—power supply usually available or made available through a generator, hand washing station with running water available in labour room; drugs—oxytocin present; infrastructure—sanctioned bed strength; service utilisation—average number of deliveries per year in 2016 (from HMIS).

We use coarsened exact matching to match PHCs in intervention and comparison groups.[39] After matching, 134 (84%) of the 159 AMANAT PHCs were matched with 56 (70%) of the 80 PHCs in the non-mentored group. From these matched PHCs, there were 265 nurses in the intervention group and 112 in the comparison group.

### Measuring nurse ability

To measure nurse ability, we developed a tool that captures both their knowledge and demonstrated skill in managing normal and complicated deliveries. Our tool adapts elements of clinical vignettes with Objective Structured Clinical Examinations (OSCEs). Clinical vignettes have been used to measure health worker knowledge in treating specific disease conditions in the context of a simulated consultation.[40–42] Healthcare providers are presented with standardised cases specific to a disease condition and their knowledge in managing the specific condition is evaluated using checklists. OSCEs are commonly used across nursing and other allied health professions and involve learners rotating through structured, time-limited stations to demonstrate discrete skills.[43 44] OSCEs are used extensively in many countries to assess competencies in obstetric and newborn care, including in India.[45–48] Our tool combined clinical vignettes and OSCEs by evaluating knowledge (via verbal responses) and skills (via demonstration) in managing normal and complicated deliveries. Nurses and ANMs were evaluated in how they would manage three standardised cases—normal birth, neonatal resuscitation and postpartum haemorrhage. These cases were selected because mentees were taught to manage these conditions; haemorrhage is a leading cause of maternal mortality, and birth asphyxia a leading cause of neonatal mortality in India.[29 30] Tool development was facilitated

by CARE programme developers, India-based obstetric experts and international experts in BEmONC training. The same tool was used in all assessments. A panel of three nurses from the USA and India (Delhi and Bihar) assessed the face and content validity (by indicating if a particular item should be included or not) of the tool.

The three cases on which nurses and ANMs were evaluated—normal delivery, postpartum haemorrhage and neonatal resuscitation—were structured to assess ability in history taking, examinations, case identification/diagnosis and management. Two interviewers administered each case—interviewer #1 played the role of the pregnant woman and wore a Mama Natalie, and interviewer #2 observed the interaction using a standard checklist. Each case began with the respondent being presented with a situation by interviewer #2—for example, the normal delivery case is introduced to the nurse/ANM as '*This is Tara [pointing to interviewer #1]. She is pregnant. She has come to see you at the clinic*'. The respondent was then asked what history questions she wanted to ask Tara, what examinations she would do, and the respondent demonstrated them on the Mama Natalie. To each relevant history question or examination done, the respondent received a standard response. For instance, if the respondent asked 'When did you have your last menstrual period (LMP)?' a standard response of '9 months' was offered or if the respondent demonstrated checking for fetal heart rate a response of '130' is given. No responses outside of the prestructured conversation were given. Based on the results of the history and examinations, the nurse was asked to state the diagnosis/clinical decision (eg, normal or complicated delivery), and then asked to use the Mama Natalie to demonstrate how she would manage the delivery. Each case comprised a set of items that corresponded to correct actions in the domains of history and examinations, case detection/diagnosis and case management. All relevant correct actions performed was marked off on a checklist.

### Data analysis

Each case comprises a set of items that correspond to correct actions necessary for managing the case. The unit of analysis is a necessary action (verbal or demonstration) provided by the respondent in the course of completing a case. Exploratory data analysis (eg, histograms) was used to examine the distribution of provider responses across PHCs and nurses. Facilities with extreme values were identified and scrutinised. We then computed the proportion of relevant correct actions in history, examination, case detection/diagnosis and management sections for each of the three cases. Next, using logistic regressions, for each case, we modelled the probability of a nurse providing a correct action as a function of mentoring status, nurse characteristics (GNM nurse (ref: ANM), age, permanent employee (ref: contractual), facility characteristics (connected to metaled road, regular power supply available or made available through a generator, hand washing station with running water available in labour room, oxytocin present; sanctioned bed strength, average number of deliveries per year in 2016, and item dummy variables. Interactions between nurse type and mentoring status were included. Regression standard errors were adjusted for clustering due to multiple observations on the same health facility. Goodness of fit test was performed on the final model by inspecting plots of predicted versus observed values. Residual plots were examined to identify outliers and influential observations. Regression results are presented in terms of marginal effects. To estimate changes in the mentoring effect over time, we took advantage of the two cross-sectional surveys on the study PHCs—first, just after mentoring was completed and a second survey some months afterwards, depending on the phase. Locally weighted regressions (lowess) were estimated in which the binary outcome of a nurse providing a correct action was regressed on time (months) since the nurse completed mentoring. In reporting study results, we followed Strengthening the Reporting of Observational Studies in Epidemiology guidelines for cross-sectional studies.[49] The statistical analysis was conducted using Stata V.14.[50]

## RESULTS

Table 1 shows the characteristics of the mentored and comparison PHCs and nurses in the matched and full sample. The full sample (see Total sample) includes the 239 mentored and comparison PHCs and 474 nurses before matching was carried out. After matching, there were 134 mentored and 56 comparison PHCs, which yielded 256 mentored and 112 comparison nurses. The matched mentored and comparison PHCs are broadly similar in their characteristics indicating that matching was successful. However, mentored PHCs conducted more deliveries on average than the comparison ones. The PHCs and nurses in the full sample (ie, Total sample in table 1), are also broadly similar to the PHCs included in the matched sample in terms of PHC and nurse characteristics, though the matched PHCs have slightly higher values on all the PHC level indicators.

There are several notable points about the characteristics of nurses. First, the overwhelming majority of nurses were ANMs; staff nurses were a minority. The mentored group had a higher proportion of ANMs relative to the comparison group. Second, the majority of nurses were regular staff (as opposed to contractual staff); their proportion is higher in the comparison group. Finally, nurses across groups are similar in average age. Nurse age is also correlated with service duration; nurses in the mentored and comparison groups had similar practice durations.

Nurses in the AMANAT programme typically experienced six full months of mentoring. Overall 93% of nurses reported having at least 6 months of mentoring exposure (table 1). Duration of mentoring ranged from 2 to 9 months, with an average of 7 months of mentoring.

**Table 1** Characteristics of matched and unmatched sample

| Indicator | Matched PHCs | | Total sample* |
|---|---|---|---|
| | **Mentored** | **Comparison** | |
| **PHCs** | | | |
| N (PHC) | 134 | 56 | 239 |
| Connected to metaled road (%) | 92 | 87 | 89 |
| Electricity always available (%) | 90 | 89 | 86 |
| Labour room has functional handwashing (%) | 100 | 100 | 93 |
| Oxytocin available (%) | 83 | 75 | 79 |
| Average number of beds | 10.9 (10.95) | 10.2 (14.44) | 15.3 (30.19) |
| Average number of deliveries in last year | 193.4 (93.91) | 133.8 (93.39) | 170.6 (101.11) |
| **Nurses** | | | |
| N (staff nurses and ANMs) | 265 | 112 | 474 |
| Completed full course of mentoring (%) | 93 | – | – |
| ANMs (%) | 89 | 87 | 84 |
| Regular (non-contractual) staff (%) | 63 | 71 | 67 |
| Age (years) | 41.1 (8.75) | 43.2 (9.70) | 41.8 (9.22) |

*(1) The 'Total sample' is the pool of mentored (159) and non-mentored (80) PHCs from where the matched sample of mentored (134) and non-mentored (56) PHCs was drawn. Consequently, because some mentored and comparison PHCs were unmatched, the Total sample does not equal the sum of matched mentored and non-mentored PHCs. (2) Figures in parenthesis are SD.
ANMs, auxiliary nurse-midwives; PHCs, primary health centres.

These results suggest that there was high adherence to the AMANAT programme among those who participated.

### ANM and staff nurse ability

In all three cases, mentored ANMs and staff nurses provided significantly more correct actions than the comparison group (table 2). For example, mentored (comparison) ANMs provided 53% (36%) correct actions for the normal delivery case, a percentage point difference of 17 points; 49% (24%) for the postpartum haemorrhage case, a percentage point difference of 29 points, and 53% (25%) for the neonatal resuscitation case, a

percentage point difference of 28 points. However, the overall proportion of correct actions among mentored ANMs and staff nurses does not exceed 60% in any case. Table 2 also shows the predicted probabilities from the regressions with respondent and PHC characteristics at their reference values. For example, for the normal delivery case mentored (comparison) ANMs provided 53% (36%) of the correct actions, while mentored staff nurses (comparison) provided 55% (42%) correct actions. Notably, the 95% CI between mentored and comparison nurses do not overlap—mentored ANMs and staff nurses had significantly higher percentage of correct actions compared with the comparison group.

The ability of individual mentored nurses varied substantially. In the normal delivery case, nurses provided between 20% and 90% of correct actions, in the postpartum haemorrhage case they provided between 11% and 89% of correct actions, and in the neonatal resuscitation case they provided between 6% and 95% of the correct actions (results not shown). Further, a substantial number of mentored nurses provided less than half the necessary correct actions required to manage these cases. For normal delivery 41%, postpartum haemorrhage 60%, and for neonatal resuscitation 45% of the nurses provided less than half the correct actions necessary for managing these cases.

Mentored ANMs and staff nurses demonstrated significantly more correct actions for history and examinations, and case management relative to those in the comparison group (table 3). Mentored (comparison) ANMs provided 54% (41%) of the correct history and examination actions, 63% (34%) for postpartum haemorrhage and 55% (28%) for neonatal resuscitation. Similarly, mentored nurses performed better than the comparison group—mentored (comparison) ANMs provided 51% (28%) of correct case management actions for normal delivery, 37% (16%) for postpartum haemorrhage and 51% (22%) for neonatal resuscitation. These patterns are also seen in the performance of staff nurses.

The majority of ANMs and staff nurses were able to correctly identify cases (except for ANMs in the comparison group for neonatal resuscitation case) (table 3). Expectedly, the percent of correct responses was highest for the normal delivery case since the preservice training of both ANMs and staff nurses include the basics of managing normal delivery. However, the ability to correctly identify a case was higher among mentored nurses (table 3). Mentored (comparison) ANMs correctly identified 96% (99%) of the normal delivery cases, 84% (51%) of the postpartum haemorrhage case and 83% (50%) of the neonatal resuscitation case. Similarly, mentored staff nurses had higher case identification than the comparison group except for the normal delivery case. In general, staff nurses did much better than ANMs in correctly identifying a particular case.

Of interest is to see how well ANMs compare with staff nurses in their ability to manage normal and complicated deliveries. Comparison group (ie, those without

**Table 2** Overall ability of staff nurse and auxiliary nurse-midwife (ANM) to manage normal and complicated deliveries

| | Normal delivery | | Postpartum haemorrhage | | Neonatal resuscitation | |
|---|---|---|---|---|---|---|
| | M | C | M | C | M | C |
| ANM | | | | | | |
| N (ANMs) | 237 | 98 | 237 | 98 | 237 | 98 |
| n (items) | 11 850 | 4493 | 4266 | 1617 | 3792 | 1438 |
| % of correct actions (unadjusted) | 53 | 36 | 49 | 24 | 53 | 25 |
| Predicted probability of correct action | 0.53 | 0.36 | 0.50 | 0.25 | 0.53 | 0.24 |
| 95% CI | (0.51 to 0.55) | (0.33 to 0.38) | (0.48 to 0.53) | (0.22 to 0.28) | (0.49 to 0.56) | (0.19 to 0.30) |
| Staff nurse | | | | | | |
| N (staff nurse) | 28 | 14 | 28 | 14 | 28 | 14 |
| n (items) | 1400 | 1108 | 504 | 399 | 448 | 354 |
| % of correct actions (unadjusted) | 55 | 41 | 48 | 33 | 58 | 36 |
| Predicted probability of correct action | 0.55 | 0.42 | 0.50 | 0.37 | 0.57 | 0.38 |
| 95% CI | (0.51 to 0.59) | (0.37 to 0.48) | (0.44 to 0.56) | (0.31 to 0.42) | (0.49 to 0.66) | (0.31 to 0.46) |

Predicted probabilities are the probability that a staff nurse/ANM provide a correct action, with all other variables in the regression at their reference value. Multiplying these figures by 100 gives the percentage of correct actions provided by the staff nurse/ANM.

C, comparison; M, mentored.

any mentoring) ANMs had significantly lower ability relative to staff nurses for all three cases. However, ANMs that experienced mentoring had similar levels of performance as staff nurses on all three cases. For example, for the normal delivery case mentored ANMs (staff nurses) provided 53% (55%) correct actions, 50% (50%) for postpartum haemorrhage and 53% (57%) for neonatal resuscitation (table 2). The overlapping CIs between mentored ANMs and staff nurses indicates that there are no statistically significant differences between them.

**Effect of mentoring**

Mentored ANMs and staff nurses had significantly greater proportion, relative to the comparison group, of correct actions overall, and for the domains of history and examinations, case identification (except for normal delivery) and in case management (tables 2 and 3). Figure 1 illustrates the adjusted mentoring effect (ie, treatment effect on the treated) on ANMs and staff nurses. These are marginal effects from the regression models with other

**Table 3** Correct actions provided by auxiliary nurse-midwives (ANMs) and staff nurses by case domains

| | Normal delivery | | Postpartum haemorrhage | | Neonatal resuscitation | |
|---|---|---|---|---|---|---|
| | M | C | M | C | M | C |
| ANM, % (items) | | | | | | |
| N (ANMs) | 237 | 98 | 237 | 98 | 237 | 98 |
| Correct history and examination | 54* (5925) | 41 (2246) | 63* (1422) | 34 (539) | 55* (474) | 28 (180) |
| Correct case identification | 96 (237) | 99 (90) | 84* (237) | 51 (90) | 83* (237) | 50 (90) |
| Correct case management actions | 51* (5688) | 28 (2156) | 37* (2607) | 16 (988) | 51* (3081) | 22 (1168) |
| Staff nurses, % (items) | | | | | | |
| N (staff nurses) | 28 | 14 | 28 | 14 | 28 | 14 |
| Correct history and examination | 55* (700) | 41 (554) | 63* (168) | 46 (133) | 48 (56) | 28 (44) |
| Correct case identification | 96 (28) | 100 (22) | 89 (28) | 76 (22) | 100* (28) | 75 (22) |
| Correct case management actions | 54* (672) | 38 (531) | 38* (308) | 22 (244) | 56* (364) | 34 (288) |

*$\chi^2$ test p value <0.05.

C, comparison; M, mentored.

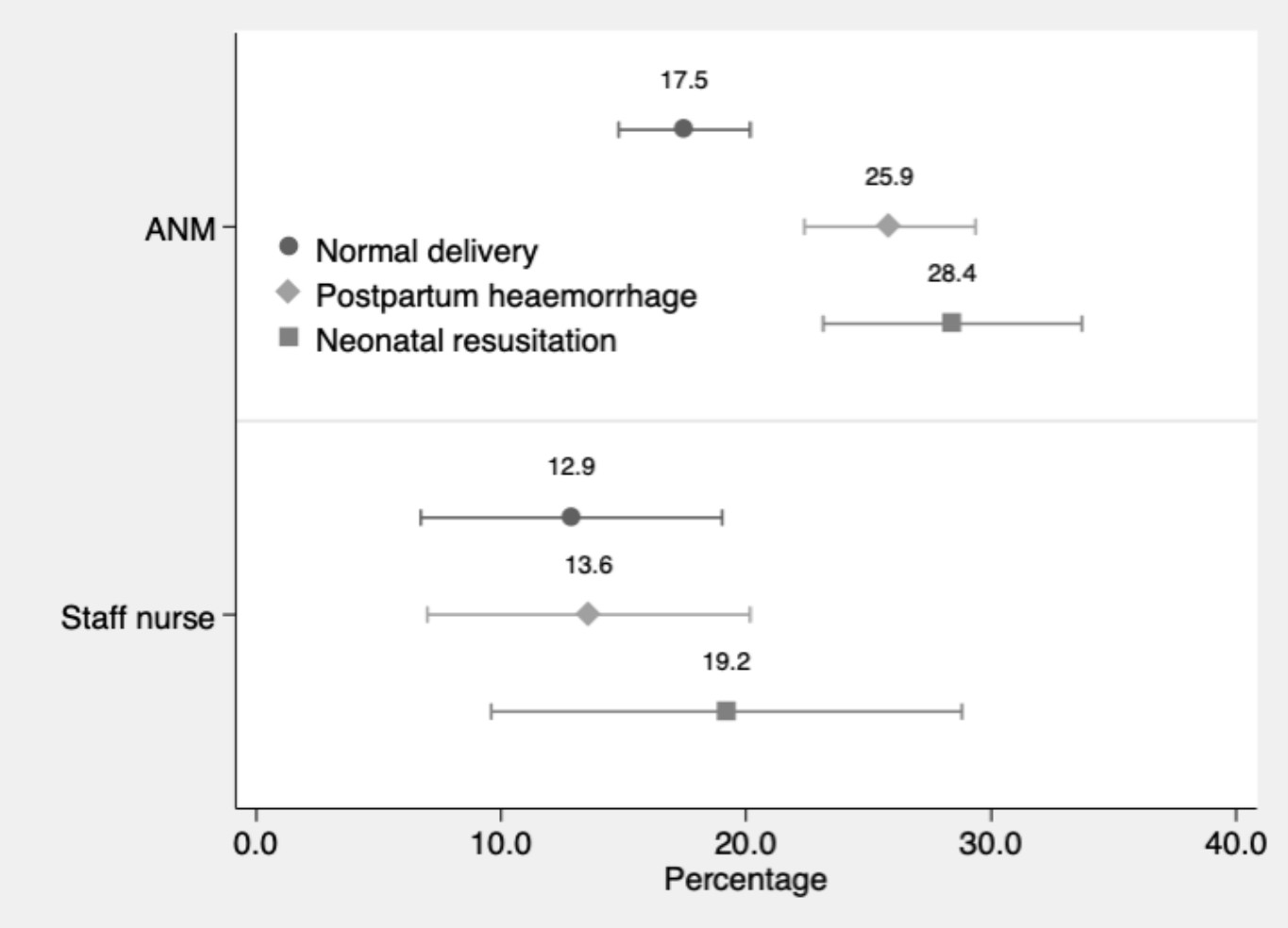

**Figure 1** Effect of mentoring on ability of ANMs and staff nurses.

variables at their reference values. ANMs in the mentored group had 17.5% more correct actions for normal delivery relative to the comparison group, 25.9% more for postpartum haemorrhage and 28.4% more for neonatal resuscitation. Staff nurses in the mentored group also show similar levels of improved performance. Notably, in all cases the mentoring effect is statistically significant, that is, the 95% CI do not overlap zero.

Both ANMs and staff nurses appear to have gained from mentoring, but ANMs have gained more. As seen in figure 1, the mentoring effect for each case is greater for ANMs than staff nurses. Finally, because no differences in the proportion of correct actions are seen between mentored ANMs and staff nurses (table 2), it appears that mentoring brought ANMs on par with staff nurses in the management of normal deliveries, postpartum haemorrhage and neonatal resuscitation.

Of interest is to know if the mentoring effect on nurse ability changes with time. Figure 2 illustrates this by plotting estimates from local area regressions within the matched sample of PHCs; it shows the probability of nurses demonstrating a correct action at different points in time after mentoring was completed. Note that the same nurses are not observed at each point in time. In all

three cases, nurse ability declined with time. In the normal delivery case, just after mentoring nurses were able to provide 53% of the correct actions, while nurses observed 43 months (the observation farthest from completion of mentoring) after mentoring were able to provide 43% of correct actions, a 10% point drop. Similar trends are seen in the case of neonatal resuscitation and postpartum haemorrhage. In the neonatal resuscitation case, the slight increase seen at the tail end appears to be an artefact of the data. For this case, nurses just after mentoring were able to provide 55% of the correct actions, which declined to 44% after 43 months postmentoring. For postpartum haemorrhage, just after mentoring nurses were able to provide 50% of correct actions, which fell to 37% 43 months postmentoring, a drop of 13% points.

## DISCUSSION

In India's public sector health system, ANMs function as community-based paramedical health workers and provide a range of services such as family planning, immunisations and antenatal care.[21 22] In human resource scarce states like Bihar, where there is a paucity of trained doctors and staff nurses, ANMs are the de facto providers

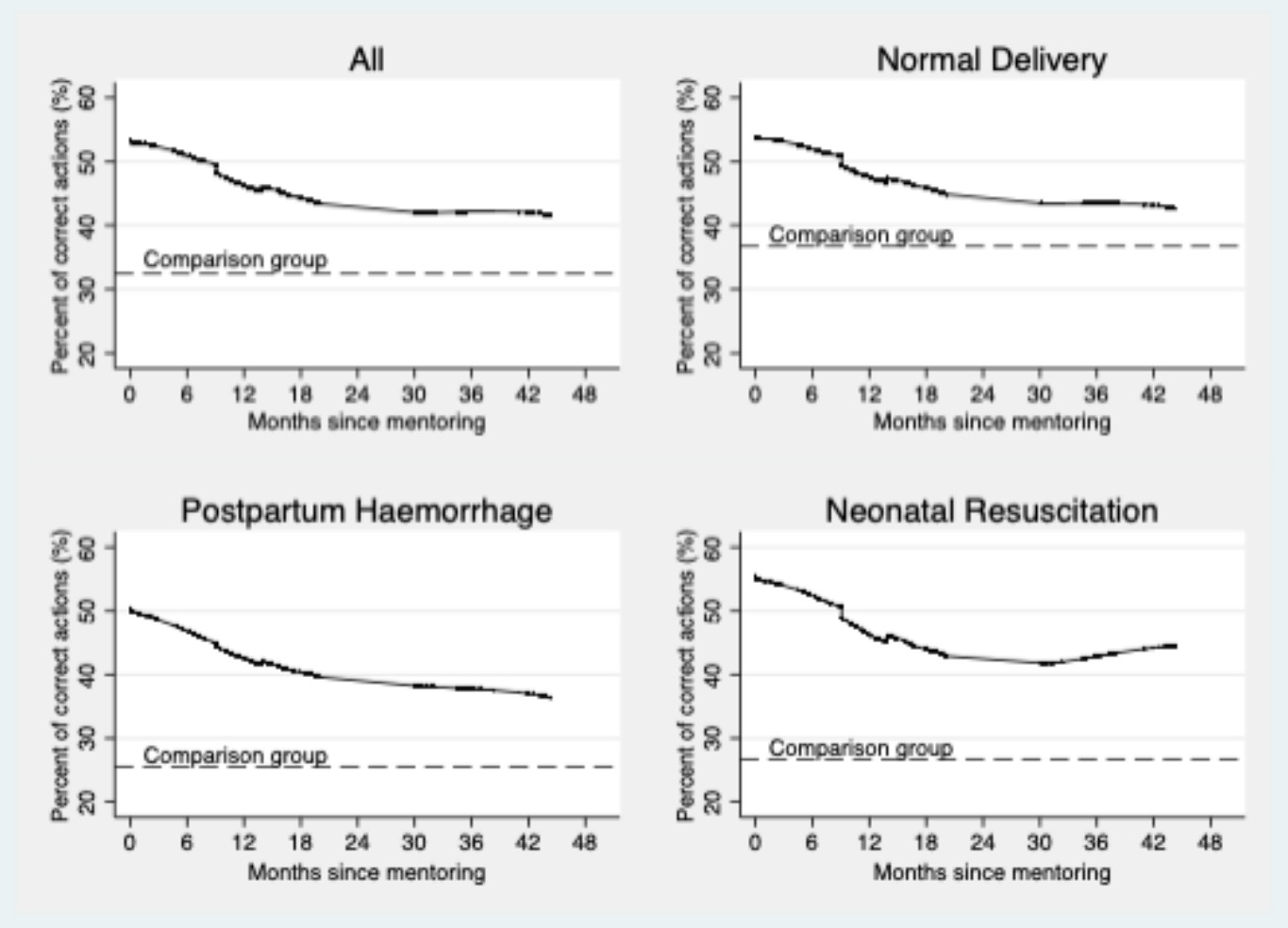

**Figure 2** Trends in nurse ability after completion of mentoring (ANM and staff nurse combined, lowess estimates).

of obstetric care at many primary care facilities even though they are not formally trained for this role. In our study, in the 239 PHCs surveyed, staff nurses comprised only 16% of the sampled nurses. In a policy environment that aggressively promotes institutional deliveries, the quality of care provided by ANMs is an important concern.

A principal finding from this study is that mentoring improved the average ability of ANMs and staff nurses at PHCs to manage normal and complicated deliveries. In all three cases—normal delivery, postpartum haemorrhage and neonatal resuscitation—mentored ANMs and staff nurses had better average ability compared with non-mentored ones. This finding is consistent with evaluations of mentoring programmes in other contexts and India.[24–27] Mentored ANMs (staff nurses) were able to provide 17% (13%) more correct actions for managing normal deliveries, 26% (14%) more for postpartum haemorrhage and 28% (19%) more for neonatal resuscitation, compared with their non-mentored colleagues. The gains were greater for ANMs since they started from a lower level of ability relative to staff nurses. Among these three cases, for both ANMs and staff nurses the largest gain in ability was in neonatal resuscitation, followed by postpartum haemorrhage, and normal delivery. This is along

expected lines since both ANMs and staff nurses are most familiar with the management of normal deliveries, so the incremental effect of mentoring would be relatively smaller. Further, the AMANAT mentoring programme emphasised management of complications such as postpartum haemorrhage and neonatal resuscitation, which explains the gains observed in this area.

ANMs that had completed mentoring had the same average ability as staff nurses to manage normal and complicated deliveries. Further, un-mentored ANMs had significantly poorer ability than un-mentored staff nurses. This mentored ANM achievement is particularly salient because ANMs typically operate as paramedical health workers providing services such as family planning, immunisations and some level of antenatal care. It is unlikely that improvements in ANM ability would be achievable outside a mentoring context. The AMANAT programme provided a combination of didactic teaching, supportive supervision and professional support over a long period of time in the context of a mentor–mentee relationship. To the best of our knowledge, there was no difference in the way staff nurses and ANMs were mentored, though the programme was focused on ANMs given that they primarily manage deliveries at PHCs.

Mentoring improved case identification among ANMs. The vast majority of mentored ANMs were able to correctly identify a normal delivery case, 84% correctly identified a postpartum haemorrhage case, and 83% the neonatal resuscitation case. While non-mentored ANMs and staff nurses performed similarly as mentored ANMs in correctly identifying normal deliveries, they did quite poorly in identifying the other two cases. For example, among non-mentored ANMs, only half of them were able to correctly identify the postpartum haemorrhage or neonatal resuscitation case. The biggest improvement in correctly identifying a case was among ANMs for postpartum haemorrhage and neonatal resuscitation. The improvements observed in recognising birth complications is important for identifying women for referral or complication management within the PHC.

Even as mentoring improved the average ability of nurses to provide quality care, there was substantial variation in nurse ability. Moreover, many mentored nurses were unable to provide a high level of correct actions. For example, 43% of the mentored nurses provided less than half the necessary actions in the normal delivery case. This suggests that participation in the mentoring programme did not raise the ability of all nurses to some minimum standard of ability. There are several reasons for this, such as, the differential capacity of individual nurses to learn, the persistence of habit or features in the work environment that prevents nurses from imbibing new learnings, or low baseline knowledge levels that impedes advanced learning. An important implication is that exposure to mentoring (or any training programme for that matter) does not automatically translate into improved ability to provide quality care for everyone. As such, it becomes important to assess how well mentored nurses provide quality care and only deploy those who meet some minimum quality standard.

There are several reasons why findings from this study might be generalisable to Bihar and to similar contexts. For one, the mentoring programme was large scale and covered most of the state. PHCs surveyed in the study covered the majority of the intervention PHCs and characteristics of the matched sample of PHCs and nurses aligns well with that of the larger pool.

It is important to be cautious about the extent to which quality of care can be improved by a single mentoring programme. Even after 6–9 months of mentoring through the AMANAT programme there remained considerable scope for improving quality of care. Mentored ANMs and staff nurses were able to provide a little more than half the correct actions on all three cases. This is also reflected in the ability domains of history and examinations and case management on which nurses were evaluated. For example, mentored ANMs (nurses) were able to provide only 51% (54%) of correct actions for case management in the normal delivery case, 37% (38%) in the postpartum haemorrhage case and 51% (56%) correct actions in the neonatal resuscitation case. Second, there will be variation in the ability of both ANMs and staff nurses post

mentoring. Therefore, not all mentored ANMs and staff nurses will have the ability to properly manage normal or complicated deliveries. Moreover, the work environment can also affect nurse performance; the availability of adequate infrastructure, adequate staff, the management environment are all factors that can influence performance and modify the effect of mentoring.

Another reason for being cautious is that the mentoring effect on nurse ability appears to decay over time. In all three cases, we find that nurse ability was highest in the period immediately after mentoring and this declined with the passage of time. The decay was least for the normal delivery case and most for the neonatal resuscitation case. This has also been observed in other nurse mentoring studies in India, which have documented declines in nurse practice and knowledge.[26 51] These patterns highlight the importance of continuing health worker education and make a strong case for mentoring or similar programmes which are embedded within the health system to be offered on a frequent basis to health workers.

A limitation of this study is that we did not observe the ability of nurses before they started mentoring. As such, our estimates of the mentoring effect will be biased if the matched comparison group nurses do not have the same average ability as those in the mentored group before they were mentored. If at baseline comparison nurses had lower ability relative to non-mentored nurse, estimates of the mentoring effect will be upwardly biased. There are several reasons why this may not be the case. One source of bias could arise from differences in nurse characteristics between mentored and comparison groups that are related to their ability. Both mentored and comparison group ANMs (or staff nurses) had the same standard preservice training and require the same qualifications to be recruited into government service. As such, it is unlikely that the extent or quality of nurse preservice training differed between groups. Further, in our regression, we control for difference in nurse and facility characteristics between groups. As such estimates of the mentoring effect are adjusted for differences in nurse characteristics such as age, years of service and employee type between mentored and non-mentored nurses.

Nurse ability can vary by the characteristics of the environment where they work. For example, studies have noted that the ability of clinical care providers varies with the economic condition of the community, that is, better-off areas have better quality providers.[52] Similarly, nurses working in a facility located in economically better areas (eg, close to urban areas) might have better ability. As such, differences in the facility location between the mentored and comparison groups can bias estimates of the mentoring effect on nurse ability. If mentored PHCs were located in more favourable areas then estimates of the mentoring effect will be upwardly biased. To address this, we created the comparison group PHCs to be as similar as possible to the mentored PHCs in terms of location and physical environment. Comparison group PHCs

were selected such that they were in close geographic proximity (from the same or adjacent administrative block) to the mentored PHCs. Further, we matched mentored and comparison PHCs on location, service readiness and PHC use criteria.

Improved nurse ability need not translate into better practice in managing deliveries. Factors such as motivation, the work environment and support from colleagues, among others, mediate the pathway between ability to practice. However, studies that have examined the association between ability and practice report that these measures are correlated, though there is a typically a gap between what clinicians know and what they do in practice.[53] In general, more able clinicians also 'do more' in practice, and ability measures the upper bound of what is possible in practice.[53]

Mentoring programmes can improve the quality of care of lesser trained health workers like ANMs to levels comparable to those of staff nurses who have higher levels of training. In health systems which struggle to recruit highly trained health workers to underserved areas, such mentoring programmes offer a way of making task shifting strategies successful. However, other considerations like programme costs and the ability to sustain such programmes for long periods are also important for scale-up. To sustain the quality gains from mentoring, it is important to expose health workers to regular continuing education programmes.

**Author affiliations**
[1]Department of International Health, Johns Hopkins University Bloomberg School of Public Health, Baltimore, Maryland, USA
[2]Division Health Economics Health Financing, Heidelberg Institute of Global Health, Medical Faculty and University Hospital, Heidelberg University, Heidelberg, Germany
[3]Johns Hopkins University School of Nursing, Baltimore, Maryland, USA
[4]Oxford Policy Management, Delhi, India
[5]CARE India Solutions for Sustainable Development, Patna, India
[6]Population, Family and Reproductive Health, Johns Hopkins University Bloomberg School of Public Health, Baltimore, Maryland, USA

**Acknowledgements** The authors would like to acknowledge Priya Nanda (BMGF, Delhi), Yamini Atmavilas (BMGF Delhi), Dr Sridhar Srikantiah (CARE-Bihar), Dr Tanmay Mahapatra (CARE-Bihar), Tom Newton-Lewis (Oxford Policy Management, Delhi).

**Contributors** The study was conceived and designed by KDR, NW and SA. Study protocols were developed by KDR, NW, SS and KM. Field work was conducted by KDR, NW, SS and KM. Data analysis was done by KDR, SS and SA. The paper was written primarily by KDR, SS and NW, with contributions from KM, AG, AD and SA.

**Funding** The study was funded by the Bill and Melinda Gates Foundation (https://www.gatesfoundation.org/) via grant number OPP1142884. The funders had no role in study design, data collection and analysis, decision to publish or preparation of the manuscript.

**Competing interests** AG and AD are employees of CARE—Bihar, the agency that designed and implemented the AMANAT programme.

**Patient consent for publication** Not required.

**Ethics approval** Ethical clearance for the study was received from the Centre for Media Studies (New Delhi) Institutional Review Board vide approval number IRB00006230.

**Provenance and peer review** Not commissioned; externally peer reviewed.

**Data sharing statement** The data contained within this study can be obtained by writing to kdrao@jhu.edu.

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
