## [Reviewer comments · BMJ Open]

ARTICLE DETAILS

TITLE (PROVISIONAL)	Where there is no nurse – an observational study of large-scale mentoring of auxiliary nurses to improve quality of intrapartum care at primary health centers in India.
AUTHORS	Rao, Krishna; Srivastava, Swati; Warren, Nicole; Mayra, Kaveri; Gore, Aboli; Das, Aritra; Ahmed, Saifuddin

VERSION 1 – REVIEW

REVIEWER	Barbara Madaj Liverpool School of Tropical Medicine, United Kingdom
REVIEW RETURNED	06-Nov-2018

GENERAL COMMENTS	Overall this is an interesting and important study on a topic which is of relevance to a wider community and as such warrants publication. However, there are a number of clarifications and amendments which are needed to make the paper clearer, which I have listed per section. Abstract - Misses information on the retention component of the study.- Strengths and limitations box: section to be reviewed and revised; currently no limitations listed; last two points seem quite neutral in message – please explain what aspects they cover. Background - Suggestion: Paragraph on Bihar as study setting may be better placed above the aim or the study?- Programme (AMANAT) and details need to be referenced or explained as they are not necessarily self-explanatory. Design - Mentoring programme background states that certain criteria were used for selecting the sites including staff availability and infrastructure – if that were the case, how can these details be used to inform the interpretations of the results?- Please explain why the three aspects of care (normal delivery, pph and newborn resuscitation) were used as focus for mentoring.- Mentoring programme requires details/clarifying: what curriculum was used for the programme? What is it adapted to local context? What language were the materials in and what language was the mentoring delivered in? were sessions following a standard agenda wrt context and topics covered or were adapted? If the latter, how? Length of time per mentoring session? Were all staff invited and how was the issue of not all staff being present at the time of the session addressed: were sessions repeated? Who were the mentors and how was the quality of their work assured? Was it always the same mentors visiting the same facilities? How
---

was the mentoring delivered: groups, pairs, individually? Please provide some details of this: what was the number of people in a group? What was the group composition – mix of ANMs and SNs? Separate groups? In terms of the implementation: how many sessions would an individual ANM/SN attend? How many ‘repeats’ of a topic would they on average do?

- The actual study design ought to be clearly stated. Then an appropriate checklist (e.g. STROBE) should be used.
- In the patient and public involvement section author mention that nurses were consulted – were these nurses in Bihar? How many? How were they selected? How was this done? Some more detail would be useful.

Methods

- Four phases of the programme are mentioned – please clarify what difference was there (if any) between phases 2,3 and 4? Why were they distinct?

- Tools used for the assessment – were they validated? Same ones used for all the assessments? Please provide some detail.

- Facility assessment – when was this information collected and how relevant would the details be to assessing the knowledge and skills; were there repeated measures re infrastructure and other ‘availability’ measures?

- Use of tablets for data collection is mentioned, but please explain how data was provided – self-administered? Tutor administered? Third party administrators? What language were the tests administered in? Were there any issues related to that?

- Were individuals followed for the retention study or cross-sectional assessment of the cohort? Any background on number of mentoring sessions.

- Details of the retention assessments are needed: was it 16 months for all participants? Why this length of time?

Results

- Comment on retention between SN and AMNs may be worth raising in the paper

Table 1 – number unclear – units or measure and reference needed; for example for N (PCH) monitored: 134, Comparison: 56 but Total is at 239, rather than 190 – please explain or amend; similarly for N (staff-nurses&ANMs): 265, 112 with a total of 474 rather than 377; other results unclear in current format, for example what does ‘Regular staff’ mean? Age – presumably that refers to the average age of participants but unclear which group – was it measured at the immediate or follow up assessment? For ‘number of beds’ and ‘Number of deliveries in past year’ two numbers are presented – one in brackets- what are they referring to? The note marked with an asterisk is unclear – please rephrase.

Discussion

- Some information used in the discussion is not presented under results – please make sure no new information from the study is added at this stage

- Statement that mentoring in the style of AMANAT is more conducive to learning needs to be supported by evidence – this is a stipulation only in the context of this paper given no comparison for other methods is provided in the study itself.

- Comment on infrastructure affecting staff ‘ability’ needs clarifying: is that the case? With data on from a SARA assessment the authors should be in a position to comment on this.

	 - There is no mention of other aspects of what constitutes an 'enabling environment' which may need mentioning: other staff and support systems, working conditions, etc, which all could impact on staff performance. - The statement 'In general, more able clinicians also 'do more' in practice...' needs to be supported by evidence. - The final statement needs refining – cost-effectiveness of interventions also needs to be taken into account in considering scale-up and that seems to be missed in this study. General points  - Spacing esp. use of brackets following % for two groups – space missing; spelling: plural of midwife is midwives not midwifes. - Grammar/style: at times statement not clear; please review the manuscript and amended as necessary. - Spellings of cadres: lower/upper case and use of hyphen for staff nurses. - Tenses: present, past v reported speech – please standardise. - Definitions: eg. long period - please be more specific and provide a definition where needed. - Under data analysis the font size is inconsistent – please amend.
--	--

REVIEWER	Dr. Kyu Kyu Than Myanmar Program Technical Advisor, Burnet Institute, Myanmar
REVIEW RETURNED	21-Dec-2018

GENERAL COMMENTS	Reviewer comment General comment: The manuscript is of high quality and has contributed to the wider knowledge of long term mentoring of health programs for community based health workers like ANMs. I would like to congratulate the authors for their hard work. The comments and suggestions given below only aim to improve the quality of the manuscript. Abstract: Objectives: Although it states in the objective that “This study examines if a long-term mentoring program improved the ability of ANMs to provide quality obstetric care”, the study also included the staff nurses from the PHC in the study. Please make sure the objective is in line with the methodology, results and discussions. Introduction: Page 4 line 35-40: Please explain the role of ANM clearly. It was never clear what their actual role was. Are they trained to conduct normal deliveries or as stated in line 37-38 “are paramedical health workers who provide a range of services such as family planning, immunizations and some level of antenatal care” and in some places they were trained to conduct normal deliveries ?? Thus please clearly state their role and responsibility within the PHC and if informal task shifting has been going on in resource limited settings please make sure you describe the setting and roles clearly about ANM and staff nurses. Methods: Page 6 line 22: It was stated that several criteria were used for matching the mentored and non-mentored PHCs. Please mentioned what the criteria are in brief ? Page 7 line 8: It was mentioned that data collectors were also trained nurses. How were they recruited and trained? Are you treating the ANM and staff nurses from the PHC on the same level? Did you do any adjustment as their basic training is different?
---

	How were the mentors selected: are they from the system or are they specifically trained by the project ? Findings: Table 1: Please clarify content of the table to be consistent and self-explanatory  - Labour room has functional hand washing: Is that the actual number or percentage? Please clarify? - Oxytocin availability: Is that the actual number or percentage? Please clarify? - Number of beds: Actual bed or average occupancy of beds per month? Please clarify? - Number of deliveries: (What is the number in the brackets)?? - Age (What is the number in the brackets)?? Figures: All combined for ANM and staff nurses??? Not very clear? Discussion: The discussion was well written. However, the number of ANMs and staff nurses may need to be described and was there any difference in mentoring of ANMs and Staff nurses as they have different backgrounds. Meaning were ANMs more obedient and compliant to the mentors than Staff nurses? Are there any qualitative findings? As the effect of mentoring program decay over time, are there any possible solutions to it for example like internal mentoring? Can you also please mentioned about the mentors recruitment and sustainability and cost aspect of the program in your discussion.
--	---

VERSION 1 – AUTHOR RESPONSE

Response to Reviewer Comments

Reviewer: 1

Reviewer Name: Barbara Madaj

Institution and Country: Liverpool School of Tropical Medicine, United Kingdom

Please state any competing interests or state 'None declared': None declared

Please leave your comments for the authors below

Overall this is an interesting and important study on a topic which is of relevance to a wider community and as such warrants publication. However, there are a number of clarifications and amendments which are needed to make the paper clearer, which I have listed per section.

Abstract

- Misses information on the retention component of the study.

Authors: We have added this in the abstract. Thanks for pointing out.

- Strengths and limitations box: section to be reviewed and revised; currently no limitations listed; last two points seem quite neutral in message – please explain what aspects they cover.

Authors: We have now explicitly included limitations to reflect those mentioned in the discussion section.

Background

- Suggestion: Paragraph on Bihar as study setting may be better placed above the aim or the study?

Authors: We considered this but did not want to leave the aim section as the very last paragraph of the section. Plus we wanted to give more prominence to the idea of mentoring than the geographic context (i.e. Bihar).

- Programme (AMANAT) and details need to be referenced or explained as they are not necessarily self-explanatory.

Authors: We have added three new references that provide more detail about the AMANAT program. These include some recently published papers as well as documents published by the implementing agency.

Design

- Mentoring programme background states that certain criteria were used for selecting the sites including staff availability and infrastructure – if that were the case, how can these details be used to inform the interpretations of the results?

Authors: We have now added the following in the text “In our analysis we include many of these selection criteria to construct a counterfactual set of health facilities.”

- Please explain why the three aspects of care (normal delivery, pph and newborn resuscitation) were used as focus for mentoring.

Authors: We have modified the statement about this (and added references) in the Analytical Methods section (under measuring nurse ability) – “These cases were selected because mentees were taught to manage these conditions; hemorrhage is a leading cause of maternal mortality, and birth asphyxia a leading cause of neonatal mortality in India.”

- Mentoring programme requires details/clarifying: what curriculum was used for the programme? Was it adapted to local context? What language were the materials in and what language was the mentoring delivered in? were sessions following a standard agenda wrt context and topics covered or were adapted? If the latter, how? Length of time per mentoring session? Were all staff invited and how was the issue of not all staff being present at the time of the session addressed: were sessions repeated? Who were the mentors and how was the quality of their work assured? Was it always the same mentors visiting the same facilities? How was the mentoring delivered: groups, pairs, individually? Please provide some details of this: what was the number of people in a group? What was the group composition – mix of ANMs and SNs? Separate groups? In terms of the implementation: how many sessions would an individual ANM/SN attend? How many ‘repeats’ of a topic would they on average do?

Authors: These are important points. We have re-done the Background section (AMANAT nurse mentoring program in Bihar) to address the questions raised here.

- The actual study design ought to be clearly stated. Then an appropriate checklist (e.g. STROBE) should be used.

Authors: We have now done this. We use the STROBE checklist for cross-sectional studies.

- In the patient and public involvement section author mention that nurses were consulted – were these nurses in Bihar? How many? How were they selected? How was this done? Some more detail would be useful.

Authors: We have added this information.

Methods

- Four phases of the programme are mentioned – please clarify what difference was there (if any) between phases 2,3 and4? Why were they distinct?

Authors: We have added “The AMANAT program was implemented in a staggered manner over four phases; there was no difference in the intervention between phases [27].”

- Tools used for the assessment – were they validated? Same ones used for all the assessments?
Please provide some detail.

Authors: We have added this “The same tool was used in all assessments. A panel of three nurses from the USA and India (Delhi and Bihar) assessed the face and content validity (by indicating if a particular item should be included or not) of the tool.”

- Facility assessment – when was this information collected and how relevant would the details be to assessing the knowledge and skills; were there repeated measures re infrastructure and other ‘availability’ measures?

Authors: We have modified the text as “Further, in 2016, as part of a larger study, a single cross-section survey of health facilities (“facility survey”) was carried out on the service delivery readiness of PHCs in Bihar. All PHCs (and other health facilities) were surveyed and information on availability and condition of drugs, supplies, equipment, and building condition. This information was used for matching intervention and comparison PHCs.”

- Use of tablets for data collection is mentioned, but please explain how data was provided – self-administered? Tutor administered? Third party administrators? What language were the tests administered in? Were there any issues related to that?

Authors: We have re-written the Data collection and questionnaires sections to incorporate this comment.

- Were individuals followed for the retention study or cross-sectional assessment of the cohort? Any background on number of mentoring sessions.

- Details of the retention assessments are needed: was it 16 months for all participants? Why this length of time?

Authors: In Data collection and questionnaires we now state that “Data for the study was collected between September 2016 and November 2017 by a series of cross-sectional surveys of PHCs. PHCs and nurses in phases 2, 3 and 4 were first observed soon after mentoring was completed and the followed-up once after that. Phase 2 PHCS were visited 3 and 15 months after mentoring was completed, Phase 3 PHCS were visited 1 and 13 months after mentoring was completed and Phase 4 PHCs were visited just after and 6 months after mentoring was completed. Data was collected using tablets by trained enumerators who had a GNM or higher nursing degree. Enumerators were recruited from other parts of India and trained in Bihar for ten days to two weeks, depending on the particular batch of enumerators, on practices of managing normal and complicated deliveries, and administering the questionnaires.”

Unfortunately, we don’t have information on the number of mentoring sessions attended. However, we have information on how many months on mentoring a mentee attended (See Table 1).

Results

- Comment on retention between SN and AMNs may be worth raising in the paper

Authors: If we understood this comment correctly, the reason why we don't comment on retention between SN and ANM is because the SN sample is thin.

Table 1 – number unclear – units or measure and reference needed; for example for N (PCH) monitored: 134, Comparison: 56 but Total is at 239, rather than 190 – please explain or amend; similarly for N (staff-nurses&ANMs): 265, 112 with a total of 474 rather than 377;

Authors: We have now added an explanation for this difference below the table “*Note: (1) *The ‘Total sample’ is the pool of mentored (159) and non-mentored (80) PHCs from where the matched sample of mentored (134) and non-mentored (56) PHCs was drawn. Consequently, because some mentored and comparison PHCs were unmatched, the Total Sample does not equal the sum of matched mentored and non-mentored PHCs. (2) Figures in parenthesis are SD..”

other results unclear in current format, for example what does ‘Regular staff’ mean? Age – presumable that refers to the average age of participants but unclear which group – was it measured at the immediate or follow up assessment? For ‘number of beds’ and ‘Number of deliveries in past year’ two numbers are presented – one in brackets- what are they referring to? The note marked with an asterisk is unclear – please rephrase.

Authors: We have added more detail to the variables in the table. The Note at the bottom of the table has also been revised (see above).

Discussion

- Some information used in the discussion is not presented under results – please make sure no new information from the study is added at this stage.

Authors: We have addressed this by modifying the text.

- Statement that mentoring in the style of AMANAT is more conducive to learning needs to be supported by evidence – this is a stipulation only in the context of this paper given no comparison for other methods is provided in the study itself.

Authors: We have deleted this sentence.

- Comment on infrastructure affecting staff 'ability' needs clarifying: is that the case? With data on from a SARA assessment the authors should be in a position to comment on this.

Authors: We have addressed this by restricting the statement to location characteristics.

- There is no mention of other aspects of what constitutes an 'enabling environment' which may need mentioning: other staff and support systems, working conditions, etc, which all could impact on staff performance.

Authors: We have added as sentence on this on page 9.

- The statement 'In general, more able clinicians also 'do more' in practice...' needs to be supported by evidence.

Authors: We have added a reference for this.

- The final statement needs refining – cost-effectiveness of interventions also needs to be taken into account in considering scale-up and that seems to be missed in this study.

Authors: We have added a sentence in the last paragraph to highlight the cost of scaling up. Our study was focused on the effectiveness of mentoring but we agree that understanding if this represents good value-for-money is equally relevant.

General points

- Spacing esp. use of brackets following % for two groups – space missing; spelling: plural of midwife is midwives not midwifes.

Authors: Thanks for pointing out – we have made changes.

- Grammar/style: at times statement not clear; please review the manuscript and amended as necessary.

Authors: We have reviewed and tried to make changes where possible.

- Spellings of cadres: lower/upper case and use of hyphen for staff nurses.

Authors: Change made for staff nurses – we removed the use of hyphen. We were not sure about the point on cadres.

- Tenses: present, past v reported speech – please standardise.

- Definitions: eg. long period - please be more specific and provide a definition where needed.

- Under data analysis the font size is inconsistent – please amend.

Authors: Thanks for pointing out – the change has been made.

Reviewer: 2

Reviewer Name: Dr. Kyu Kyu Than

Institution and Country: Myanmar Program Technical Advisor, Burnet Institute, Myanmar

Please state any competing interests or state 'None declared': Non declared

Please leave your comments for the authors below

Reviewer comment

General comment: The manuscript is of high quality and has contributed to the wider knowledge of long term mentoring of health programs for community based health workers like ANMs. I would like to congratulate the authors for their hard work. The comments and suggestions given below only aim to improve the quality of the manuscript.

Abstract:

Objectives: Although it states in the objective that “This study examines if a long-term mentoring program improved the ability of ANMs to provide quality obstetric care”, the study also included the staff nurses from the PHC in the study. Please make sure the objective is in line with the methodology, results and discussions.

Authors: we have modified the abstract to include staff nurses

Introduction:

Page 4 line 35-40: Please explain the role of ANM clearly. It was never clear what their actual role was. Are they trained to conduct normal deliveries or as stated in line 37-38 “are paramedical health workers who provide a range of services such as family planning, immunizations and some level of antenatal care” and in some places they were trained to conduct normal deliveries ?? Thus please clearly state their role and responsibility within the PHC and if informal task shifting has been going on in resource limited settings please make sure you describe the setting and roles clearly about ANM and staff nurses.

Authors: We have made this change to clarify roles of ANMs and staff nurses.

Methods:

Page 6 line 22: It was stated that several criteria were used for matching the mentored and non-mentored PHCs. Please mentioned what the criteria are in brief ?

Authors: We have added a reference to the Analytical Methods section where the matching criteria are described.

Page 7 line 8: It was mentioned that data collectors were also trained nurses. How were they recruited and trained?

Authors: We have added this in the text. “Data was collected using tablets by trained enumerated who had a GNM or higher nursing degree. These data collectors were recruited from other parts of India and trained in Bihar for ten days to two weeks, depending on the particular batch, on practices of managing normal and complicated deliveries, and administering the questionnaires.”

Are you treating the ANM and staff nurses from the PHC on the same level? Did you do any adjustment as their basic training is different?

Authors: Yes, we are treating them on the same level for managing normal and complicated deliveries because this the function that both perform. Their pre-service training is different, which is why we are interested in knowing if they perform similarly after they are exposed to mentoring.

How were the mentors selected: are they from the system or are they specifically trained by the project ?

Authors: We have added this information in the text (See AMANAT nurse mentoring program in Bihar): “The nurse-mentors were hired by CARE-Bihar, were recruited from across India, and had a BSc degree in nursing. The nurse-mentors were trained by CARE and provided six days of training on simulation facilitation, team building, communication skills, and debriefing skills, followed by a four-day refresher training around three months into the mentoring period [29].”

Findings:

Table 1: Please clarify content of the table to be consistent and self-explanatory

- Labour room has functional hand washing: Is that the actual number or percentage? Please clarify? Oxytocin availability: Is that the actual number or percentage? Please clarify? Number of beds: Actual bed or average occupancy of beds per month? Please clarify? Number of deliveries: (What is the number in the brackets)?? Age (What is the number in the brackets)??

Authors: We have made the change.

-

Figures: All combined for ANM and staff nurses??? Not very clear?

Authors: Yes, this is combined for ANM and staff nurses. We have added a clarification in the title.

Discussion: The discussion was well written. However, the number of ANMs and staff nurses may need to be described and was there any difference in mentoring of ANMs and Staff nurses as they have different backgrounds. Meaning were ANMs more obedient and compliant to the mentors than Staff nurses? Are there any qualitative findings?

Authors: To the best of our knowledge there was no difference in mentoring between staff nurses and ANMs. We did not collect information on this. However, since the vast majority of mentees were ANMs, the program was focused on them. We think this is an important point and have mentioned this in the discussion.

As the effect of mentoring program decay over time, are there any possible solutions to it for example like internal mentoring?

Authors: This is an important point and we now mention it in the discussion.

Can you also please mentioned about the mentors recruitment and sustainability and cost aspect of the program in your discussion.

Authors. We have added a sentence on the importance of costs in the last paragraph. However, we do not have any information on how much the program cost.

VERSION 2 – REVIEW

REVIEWER	Barbara Madaj Liverpool School of Tropical Medicine, United Kingdom
REVIEW RETURNED	19-Feb-2019

GENERAL COMMENTS	Thank you for providing a revised manuscript and incorporating the feedback shared. It is clear substantial work has gone into the reworking of the paper and I hope the authors also see that as a result it has become stronger. I have no issues to raise, other than some minor stylistic points for consideration:  - P. 3, lines 49- 50: 'Because the nomenclature of primary care facilities is somewhat ambiguous in Bihar and change over time [...] – should be rephrased to either changes overs time or has changes over time to make the sentence grammatically correct - P. 4, lines 16-17: 'the nurse-mentors were trained by CARE and provided six days of training [...] – should be changes to 'provided with' to indicate they were the recipients of the training rather than acting as trainers - P.5, line 26: full stop missing at the end of the sentence – please add - P.5, line 50: Phase 2 PHCS should be changes to Phase 2 PHCs - P.7, lines 19-21 is a repeated sentence from p.3 – please keep in one place only All other suggestions made in the previous review have been addressed sufficiently and I have no further comments. However, a statistical review of the methodology would be advisable.
---

REVIEWER	Kyu Kyu Than Senior Technical Advisor, Burnet Institute, Myanmar
REVIEW RETURNED	22-Feb-2019

GENERAL COMMENTS	The authors have fully address my concerns and I am satisfy with the revisions.
---

REVIEWER	Shijun Zhu University of Maryland Baltimore
REVIEW RETURNED	24-Apr-2019

GENERAL COMMENTS	Thanks for the invited review on this manuscript. It's well written/revised and I enjoyed reading it. The comments below might be helpful for further clarify the methods.  1. In Data analysis (page 8), "Regression standard errors were adjusted for clustering" please clarify the clustering on facility or repeated measures. It seems possible that an AMN been interviewed twice. 2. How the follow-up survey was conducted (timing and repeated measures?) and the related analysis on outcomes along time (E.g., Figure 2) are unclear.
--

VERSION 2 – AUTHOR RESPONSE

Reviewer Comments

1. P. 3, lines 49- 50: ‘Because the nomenclature of primary care facilities is somewhat ambiguous in Bihar and change over time [...] – should be rephrased to either changes over time or has changes over time to make the sentence grammatically correct

Authors: Change made

2. P. 4, lines 16-17: ‘the nurse-mentors were trained by CARE and provided six days of training [...] – should be changes to ‘provided with’ to indicate they were the recipients of the training rather than acting as trainers

Authors: Change made

3. P.5, line 26: full stop missing at the end of the sentence – please add

Authors: Change made

4. P.5, line 50: Phase 2 PHCS should be changes to Phase 2 PHCs

Authors: Change made

5. P.7, lines 19-21 is a repeated sentence from p.3 – please keep in one place only

Authors: Could not find this sentence. It seems the line numbers don't match up.

6. In Data analysis (page 8), “Regression standard errors were adjusted for clustering” please clarify the clustering on facility or repeated measures. It seems possible that an AMN been interviewed twice.

Authors: We have now clarified that clustering was at the facility level.

7. How the follow-up survey was conducted (timing and repeated measures?) and the related analysis on outcomes along time (E.g., Figure 2) are unclear.

Authors: In the Study Design section we had mentioned the multiple observations made on the same facility. We have now added the following in the Data analysis section – “To estimate changes in the mentoring effect over time we took advantage of the two cross-sectional surveys on the study PHCs - first, just after mentoring was completed and a second survey some months afterwards, depending on the phase. Locally weighted regressions (lowess) were estimated in which the binary outcome of a nurse providing a correct action was regressed on time (months) since the nurse completed mentoring.”